# Spatio-Temporal Variability of Malaria Incidence in the Health District of Kati, Mali, 2015–2019

**DOI:** 10.3390/ijerph192114361

**Published:** 2022-11-02

**Authors:** Abdoulaye Katile, Issaka Sagara, Mady Cissoko, Cedric Stephane Bationo, Mathias Dolo, Ismaila Thera, Siriman Traore, Mamady Kone, Pascal Dembele, Djoouro Bocoum, Ibrahima Sidibe, Ismael Simaga, Mahamadou Soumana Sissoko, Jordi Landier, Jean Gaudart

**Affiliations:** 1INSERM, IRD, SESSTIM, ISSPAM, UMR1252, Faculty of Medicine, Aix Marseille University, 13005 Marseille, France; 2Malaria Research and Training Center (MRTC), FMOS-FAPH, Mali-NIAID-ICER, Université des Sciences, des Techniques et des Technologies de Bamako, Bamako BP 423, Mali; 3Programme National de Lutte Contre le Paludisme, Bamako BP 233, Mali; 4Direction Nationale de L’Hydraulique, Bamako BP 66, Mali; 5Centre de Santé de Référence du District Sanitaire de Kati, Région de Koulikoro, Kati BP 594, Mali; 6APHM, INSERM, IRD, SESSTIM, ISSPAM, UMR1252, Hop Timone, BioSTIC, Biostatistic & ICT, Faculty of Medicine, Aix Marseille University, 13005 Marseille, France

**Keywords:** malaria, environmental factors, hotspot, spatio-temporal dynamics, geoepidemiology

## Abstract

Introduction: Despite the implementation of control strategies at the national scale, the malaria burden remains high in Mali, with more than 2.8 million cases reported in 2019. In this context, a new approach is needed, which accounts for the spatio-temporal variability of malaria transmission at the local scale. This study aimed to describe the spatio-temporal variability of malaria incidence and the associated meteorological and environmental factors in the health district of Kati, Mali. Methods: Daily malaria cases were collected from the consultation records of the 35 health areas of Kati’s health district, for the period 2015–2019. Data on rainfall, relative humidity, temperature, wind speed, the normalized difference vegetation index, air pressure, and land use–land cover were extracted from open-access remote sensing sources, while data on the Niger River’s height and flow were obtained from the National Department of Hydraulics. To reduce the dimension and account for collinearity, strongly correlated meteorological and environmental variables were combined into synthetic indicators (SI), using a principal component analysis. A generalized additive model was built to determine the lag and the relationship between the main SIs and malaria incidence. The transmission periods were determined using a change-point analysis. High-risk clusters (hotspots) were detected using the SatScan method and were ranked according to risk level, using a classification and regression tree analysis. Results: The peak of the malaria incidence generally occurred in October. Peak incidence decreased from 60 cases per 1000 person–weeks in 2015, to 27 cases per 1000 person–weeks in 2019. The relationship between the first SI (river flow and height, relative humidity, and rainfall) and malaria incidence was positive and almost linear. A non-linear relationship was found between the second SI (air pressure and temperature) and malaria incidence. Two transmission periods were determined per year: a low transmission period from January to July—corresponding to a persisting transmission during the dry season—and a high transmission period from July to December. The spatial distribution of malaria hotspots varied according to the transmission period. Discussion: Our study confirmed the important variability of malaria incidence and found malaria transmission to be associated with several meteorological and environmental factors in the Kati district. The persistence of malaria during the dry season and the spatio-temporal variability of malaria hotspots reinforce the need for innovative and targeted strategies.

## 1. Introduction

Malaria continues to be a serious threat to populations living in endemic or epidemic areas. According to the World Health Organization (WHO), the global number of malaria cases in 2019 was 229 million, with 94% of these occurring in sub-Saharan Africa [1]. Of the 409,000 malaria deaths reported in 2019, 67% occurred in children under five years and 95% were recorded in sub-Saharan Africa [1]. Unfortunately, the global decrease in malaria incidence over the 2015–2019 period was negligible, at 2% [1].

In Mali, 2.9 million cases of malaria were recorded in 2019, including 871,274 severe cases and 1454 deaths [2]. That same year, malaria was one of the leading causes of mortality (27%), morbidity (23%), and consultations (34%) [2,3]. Malaria transmission is highly heterogenous in Mali, showing (i) an endemic profile (hyperendemic or holoendemic) in the Sudano–Sahelian zone (center and southern zone), where 50 to 80% of children under five years are parasite carriers (i.e., plasmodium index); (ii) an epidemic profile in the northern Saharian zone, with a plasmodium index below 5%; (iii) a hyperendemic profile (bi- or pluri-modal) in irrigated/flood-prone areas (mainly bi-annual rice culture); and (iv) a hypoendemic profile in urban areas [4]. While the spatio-temporal variability of malaria transmission in Mali is largely related to environmental and meteorological factors [3], it is also increasingly linked to climate change, as in other regions of the world [3,5].

Over the last two decades, the National Malaria Control Program (NMCP) has been implementing malaria control strategies in Mali, based on WHO recommendations. These strategies consist of early diagnosis and case management, chemoprevention in pregnant women and children aged 3 to 59 months, and vector control using long-lasting insecticide nets (LLIN) and indoor residual spraying (IRS) [2].

In Mali, early diagnosis is currently performed using rapid diagnostic tests (RDTs) and early case management through the administration of artemisinin-based combination therapy (ACT). Both RDTs and ACT are free of charge in health facilities for children under five years. According to the Malaria Indicator Survey, only 14% of febrile children in Mali received an RDT and 51% failed to take ACT in 2015 [6]. In 2021, 23.3% of children received an RDT, 19.4% of whom were positive for malaria [7].

Chemoprevention in pregnant women is based on intermittent preventive treatment (IPT), which consists of the administration of three doses of sulfadoxine-pyrimethamine (SP) during pregnancy. Coverage has considerably improved since IPT was first introduced in Mali, in 2003 [8]. Thus, the rate of coverage increased from 35% in 2012 to 66% in 2015 for the first dose, and from 44% in 2015 to 56% in 2019, and to 57% in 2021 for the second dose [2,6]. Despite this significant increase, Mali has not reached the goal of universal coverage recommended by the WHO [3], nor has it even achieved the coverage target of 80% set in the 2018–2022 National Malaria Control Strategic Plan [6]. The low coverage rate is largely explained by poor adherence to treatment among pregnant women. 

Since 2012, the WHO has recommended the administration of seasonal malaria chemoprevention (SMC) in children under five years to prevent malaria morbidity and mortality. A maximum of four doses of antimalarial treatment, a combination of SP and amodiaquine (AQ), should be administered at one-month intervals during the high transmission season [9]. According to the Malaria Indicator Survey, 36% of the targeted children in Mali received at least one dose of antimalarial treatment during the high transmission season, in 2015 [6]. Between 2016 and 2019, this rate increased significantly to 106% of the targeted children for the first and second doses, to 107% for the third dose, and to 105% for the fourth dose [2]. In spite of these significant improvements, SMC coverage is likely to have been hindered by the limited access to care among the Sahelian children and by low adherence to treatment after the first dose.

Vector control consists in preventing human contact with the mosquito vector and in limiting mosquito proliferation in the environment. This strategy specifically relies on the appropriate use of LLINs and IRS. In Mali, LLINs are distributed free of charge, both during prenatal consultations and during the distribution campaigns organized by the NMCP and its partners. According to the Malaria Indicator Survey, 71% of children under five years, 78% of pregnant women, and 68% of the population who are at risk of malaria used LLINs in Mali, in 2015 [6]. While these figures are encouraging, Mali has yet to reach the goal of universal LLIN coverage, recommended by the WHO [10]. To date, IRS has been applied in only four health districts of Mali, protecting 90% of the population at risk in these areas [3]. 

In the health district of Kati, where malaria incidence was estimated at 68% in 2017 [11], the same control strategies are implemented as in the rest of Mali. However, these strategies have not been as successful as one would have hoped. While the RDT coverage rate in Kati was high, at 81%, in 2016, it had significantly fallen to 68% in 2017 [11]. The drop in antenatal visits from 37% in 2016, to 26% in 2017, and to 24% in 2018 suggests that the IPT coverage rate in pregnant women also decreased [9]. As for the LLIN coverage rate in pregnant women, it fell from 84.2% in 2017, to 51.68% in 2018 [11]. By contrast, SMC, which was introduced in the Kati district in 2016, has shown encouraging results, with a coverage rate of 109% of the targeted children in 2019 [11]. Note that no mass IRS campaign has been implemented in Kati, to date.

Despite the implementation of control strategies at the national scale, the malaria burden remains high in Mali. In this context, a new approach is needed, which accounts for the spatio-temporal variability of malaria transmission at the local scale. To help achieve this objective, the present study sought to answer the following question: “What meteorological and environmental factors influence the spatio-temporal variability of malaria transmission in the health district of Kati, in Mali?”

## 2. Methods

### 2.1. Study Site

The health district of Kati is located north-west of Bamako, in the Sudano–Sahelian zone of Mali. In 2019, the population of Kati was estimated at 695,921, with a density of 66 inhabitants/km^2^, and a growth rate of 3.6%. The district has an area of 9636 km^2^ and is subdivided into 23 communes and 35 health areas. It is crossed by various seasonal streams and by the Niger River, in the south-east. The vegetation consists mainly of grassy savannah, dotted with fruit trees and shrubs. The main economic activities are market gardening and cereal cultivation, both of which are facilitated by the presence of water reservoirs and seasonal streams, which constitute potential mosquito breeding sites. Livestock breeding is another important economic activity, with the commune of Kati hosting the country’s largest weekly livestock fair. 

Annual rainfall in the Kati health district is 1000 mm. The rainy season runs from June to October and peak rainfall occurs in August (304 mm). The temperature averages 20 °C during the cold season and 30 °C during the hot season.

Malaria transmission in Kati is moderate [12] and seasonal, with the transmission season running from June to December. In some areas, the entomological inoculation rate can range from 137 to 167 infecting bites per person during the transmission season [13].

### 2.2. Data Collection

This study retrospectively analyzed daily malaria cases recorded in the health district of Kati, over the period 2015–2019. Malaria data were collected from the consultation records of the district’s 35 health areas. These records contained the following information on patients visiting community health centers (CSCOM): date of consultation, name, age, sex, place of residence, weight, clinical signs, diagnosis, and treatment prescribed. Only cases confirmed by thick blood smear or RDT were considered for analysis. 

Meteorological data were collected from different sources. Data generated through remote sensing were extracted from the NASA Giovanni website [14], as follows: daily number of rainfall events and cumulative rainfall (mm); minimum and maximum day and night temperatures (°C); and minimum and maximum day and night relative humidity (%). Other data were extracted from the ERA5 database through Google Earth Engine [15]: mean air pressure (hPa); normalized difference vegetation index; mean wind speed (km/h). The following hydrological data were obtained from the National Directorate of Hydraulics of Mali: Niger River height (cm) and flow (m^3^). 

The following land cover data were extracted from the Copernicus Global Land Cover [16,17,18,19,20]: bare soil, cropland, grass vegetation, shrubland, forest, seasonal inland water, and permanent inland water. 

The geographic coordinates of the 35 health areas of Kati were extracted from the Geographic Location System during the field survey. The perimeter of each health area was delineated based on these geographic coordinates, using the Voronoi polygon method.

Malaria, meteorological, and environmental data were aggregated on a weekly basis. 

### 2.3. Statistical Analysis

#### 2.3.1. Temporal Analysis of Malaria Incidence and Associated Meteorological and Environmental Factors

An additive decomposition of the malaria incidence time series was performed to estimate trend, seasonality, and residual at the scale of the Kati health district.

Weekly malaria incidence was estimated and plotted. Rainfall, mean temperature, and Niger River height and flow were also plotted at health-district level. 

To reduce dimension and, thus, account for collinearity, a principal component analysis (PCA) of meteorological and environmental factors was performed. Components were determined using the elbow method and the Kaiser criterion [21]. Each component represented a synthetic indicator (SI).

A univariate generalized additive model (GAM), using minimum generalized cross validation and maximum explained deviance, was performed to estimate the lags between the main SIs and malaria incidence [21]. 

A multivariate GAM, with negative binomial distribution, was built to account for overdispersion [22]. Spline function smoothing was performed to estimate non-linear relationships, and standardized incidence ratios were estimated by modeling the log-transformed population as the offset [23].

A change-point analysis of the malaria incidence time series was performed to determine transmission periods. The dates of significant changes in the mean and variance of malaria incidence were determined using the pruned exact linear time algorithm [24], with the AIC criterion [5,25,26].

#### 2.3.2. Spatial Analysis of Malaria Incidence

The search for high-risk clusters or hotspots was carried out using Kulldorff’s SatScan method, the precision of which increases with the size of the population at risk, the incidence rate, and the relative risk (RR) [27,28]. This method was selected because it circumvents the problem of test multiplicity by using the likelihood ratio test and the Monte Carlo approach [26]. Health areas with significantly high malaria incidence were grouped into high-risk clusters, in the form of a circular window of variable size [26], a shape that helped to avoid the poor performance associated with edge effects [29]. The characteristics of each hotspot were studied using supervised classification to identify the main associated factors and their combinations.

A map showing the spatio-temporal distribution of malaria incidence and hotspots was built for each year of the study. 

#### 2.3.3. Classification and Regression Tree Analysis

Health areas were ranked according to their level of malaria risk using classification and regression tree analysis (CART), a non-parametric classification method that analyzes historical data by means of a decision tree. The main variables associated with hotspot status were identified by constructing a tree whose leaves (i.e., the terminal nodes) were composed of health areas presenting the same level of malaria risk [30]. The CART method presents two main advantages: it can treat both numerical and categorical variables and it handles outliers more effectively than other statistical models (including PCA) [31].

#### 2.3.4. Software

Malaria data were entered on tablets using REDCap software version 11.1.1 (Vandrbilt University, Nashville, TN, USA). Meteorological and environmental data extraction and statistical analyses were performed using R software, version 4.1.0 (18 May 2021, Copyright © 2022, R Foundation for Statistical Computing). SatScan, version 9.7 (Information management Services Inc, Calverton, MS, USA), was used to detect malaria hotspots. The incidence map with malaria hotspots was built with QGIS, version 3.16.7 (Open Source Geospatial Foundation Project, Beaverton, OR, USA). Figures and images were formatted and treated using Microsoft Paint.

## 3. Results

### 3.1. General Description of the Malaria Incidence Time Series

A total of 293,001 confirmed malaria cases, which were collected from 190 consultation records, were analyzed for the period 2015–2019. 

The malaria incidence time series confirmed the seasonality of malaria transmission in Kati. The peak of malaria incidence generally occurred in October. The peak of incidence decreased over the study period, from 60 cases per 1000 person–weeks in 2015, to 27 cases per 1000 person–weeks in 2019 (Figure 1). 

The peak rainfall was observed in August in 2015, 2018, and 2019, and in July in 2016 and 2018. This peak also decreased over the study period, from 146 mm in 2015, to 105 mm in 2019 (Figure 1).

The lag between the peak rainfall and the peak malaria incidence was 8 weeks in 2015, 2018, and 2019; 13 weeks in 2016; and 14 weeks in 2017. 

Three SIs were determined using PCA, with the elbow method and the Keiser criterion. These SIs explained 88.5% of the total inertia (Figure 2). The first SI (SI#1), which explained 49.3% of the inertia, consisted of river flow and height, relative humidity, and rainfall. The second SI (SI#2), which explained 29.8% of the inertia, consisted of air pressure and temperature (Table 1). 

### 3.2. Temporal Analysis of Malaria Incidence and Associated Meteorological and Environmental Factors

The univariate GAM showed a lag of 4 weeks between SI#1 (river flow and height, relative humidity, and rainfall) and malaria incidence. A lag of 19 weeks was found between SI#2 (air pressure and temperature) and malaria incidence. 

The multivariate GAM highlighted a significant relationship between the first two SIs (SI#1 and SI#2) and malaria incidence, with an explained deviance of 78.5%. 

The relationship between SI#1 (river flow and height, relative humidity, and rainfall) and malaria incidence (*p* < 0.001) was positive and almost linear (Figure 3A). The SI#1 variables that most correlated with malaria incidence were river flow and height, and relative humidity (Table 1). 

The relationship between SI#2 (air pressure and temperature) and malaria incidence was significantly non-linear (*p* < 0.001). The SI#2 variable that most correlated with malaria incidence was temperature (Table 1): the incidence of malaria increased significantly as the temperature increased, but decreased significantly at higher temperatures (Figure 3B).

Two transmission periods were determined using a change-point analysis, as follows: a low transmission period (LTP) and a high transmission period (HTP).

The mean duration of the LTP was 24 weeks. The LPT generally started in December and ended in July; however, it lasted from December to June in 2016–2017 and from November to July in 2017–2018. 

The mean duration of the HTP was also 24 weeks. The HTP started in July and ended in December during every year of the study. 

### 3.3. Spatial Analysis of Malaria Incidence

According to the SatScan analysis, 25 out of 35 health areas were a malaria hotspot at least once during the study period. Specifically, three health areas were a malaria hotspot in the first year of the study, two health areas were a malaria hotspot in two consecutive years (2015–2016 or 2016–2017), 10 health areas were a malaria hotspot every year of the study, and 10 health areas were a hotspot without temporal regularity (Table 2). 

Malaria hotspots were detected during all LTPs.

A total of seven significant hotspots, covering nine health areas, were detected during the 2015–2016 LTP. The hotspot with the highest malaria RR (7.23; *p* < 0.0001) had a malaria incidence rate of 0.79 cases per 1000 person–weeks. This hotspot covered two health areas with a population of 10,026 inhabitants, within a radius of 11.15 km. 

A total of five significant hotspots, covering 16 health areas, were detected during the 2016 HTP. The largest hotspot had an RR of 3.87 (*p* < 0.0001) and covered nine health areas with a population of 106,485, within a radius of 29.75 km.

A total of five significant hotspots, covering eight health areas, were detected during the 2017–2018 LTP. The hotspot with the highest RR (13.45; *p* < 0.0001) covered two health areas with a population of 8195 inhabitants, within a radius of 3.84 km. 

Lastly, a total of seven significant hotspots, covering 13 health areas, were detected during the 2018 HTP. The largest hotspot had an RR of 3.17 (*p* = 0.0001) and covered six health areas with a population of 95,226 inhabitants, within a radius of 22.86 km. The hotspot with the highest RR (18.84; *p* = 0.0001) covered only one health area, with a population of 4423 inhabitants. 

Table 3 describes the transmission periods according to the presence of hotspots and according to selected meteorological variables. Table 4 describes the health areas according to the number of times they were a hotspot and according to their land cover characteristics. Maps showing the spatio-temporal distribution of malaria incidence and hotspots in the Kati health district are presented in Figure 4.

### 3.4. Classification and Regression Tree Analysis

The CART method identified three classes of malaria risk: low, moderate, and high. Two variables were significantly associated with the different classes of risk: cropland cover and bare soil cover. 

A total of 18 health areas were classified as having a low level of malaria risk (Class 1). These health areas had a mean cropland cover ≤ 23.25. The number of times these health areas were identified as a hotspot was less than one.

A high level of malaria risk (Class 2) was observed in nine health areas, which had a mean cropland cover > 23.25 and a mean bare soil cover ≤ 0.98. The number of times these health areas were identified as a hotspot was equal to four.

A total of eight health areas were classified as having a moderate level of malaria risk (Class 3). These health areas had a mean cropland cover > 23.25 and a mean bare soil cover > 0.98. The number of times these health areas were identified as a hotspot was equal to two.

The different classes of malaria risk are described in Table 5.

Figure 5 shows the CART classification of the health areas, according to the level of malaria risk and to the cropland cover and bare soil cover. A map of the health areas, classified according to the level of malaria risk, is presented in Figure 6.

## 4. Discussion

This study aimed to describe the spatio-temporal variability of malaria incidence and the associated meteorological and environmental factors, in the health district of Kati, Mali, for the period 2015–2019. Our descriptive analysis of the incidence curve confirmed the seasonality of malaria transmission in the Kati district. The spatio-temporal variability of malaria incidence was high and was associated with several meteorological and environmental factors. Malaria hotspots were detected during all LTPs.

For most of the study period, the onset of the HTP occurred in July in the Kati health district—i.e., one month later than is usually observed in this, and other health districts of the Sudano–Sahelian zone. This finding was similar to that reported by Dieng et al. in Senegal [5], which can most likely be explained by the climatic similarities between the two study sites. However, it differs from that reported by Cissoko et al. in northern Mali (Diré health district) [21], who found that the HTP onset occurred in August.

Malaria incidence decreased gradually over the study period. This decrease may be due to the following factors: (1) the impact of SMC, which was introduced in the Kati district in 2016 [32], and that was likely to have helped to reinforce the malaria control strategies already in place; (2) the observed decrease in rainfall over the study period; or (3) a combination of the two. Note that a decrease in malaria incidence has been reported in several other regions of sub-Saharan Africa and in other parts of the world in recent years [15,25,26,33,34,35]. 

The lag between the peak rainfall and the peak malaria incidence in the Kati district was 8 weeks in 2015, 2018, and 2019; 13 weeks in 2016; and 14 weeks in 2017. Similarly, the studies by Ferrão et al. in Mozambique [36] and Diouf et al. in Senegal [37] reported a lag of 8 weeks between the peak rainfall and the peak of malaria incidence, while that by Teklehaimanot et al. in Ethiopia [38] found a lag of 12 weeks between the two events. 

A lag of 4 weeks was observed between SI#1 (river flow and height, relative humidity, and rainfall) and malaria incidence. By contrast, Sissoko et al., in Mali [25], and Rouamba et al., in Burkina Faso [33], reported a lag of 8 and 9 weeks, respectively, between relative humidity and malaria incidence. Ouedraogo et al., in Burkina Faso [26], found a shorter lag of 2 weeks between these two variables. 

The relationship between SI#1 (river flow and height, relative humidity, and rainfall) and malaria incidence was positive and almost linear. Interestingly, the relationship between relative humidity and malaria incidence was also positive and almost linear in the studies by Segun et al., in Nigeria [39], and by Ouedraogo et al., in Burkina Faso [26]. Moreover, a positive and almost linear relationship between river height and malaria incidence was found in the studies by Stefani et al., in French Guiana [34], and by Sissoko et al. [25] and Cissoko et al. in Mali [21]. In our study, the correlation between rainfall and malaria incidence was not as strong as that between the other SI#1 variables and malaria incidence. This suggests that the impact of rainfall on malaria incidence in the Kati district is always mediated by relative humidity and by river flow and height [5]. 

The relationship between SI#2 (air pressure and temperature) and malaria incidence was non-linear, with a lag of 19 weeks. The incidence of malaria increased gradually as the temperature increased but decreased significantly at higher temperatures—a phenomenon explained by the fact that *Anopheles gambiae* (the principal vector for this area) larvae and adults die at temperatures above 35 °C [40]. A non-linear relationship between temperature and malaria incidence was also observed in the studies by Sissoko et al., in Mali [25], and by Ouedraogo et al. [26] and Bationo et al., in Burkina Faso [15]. However, the lag between temperature and malaria incidence in these studies was 13, 2, and 16 weeks, respectively.

The LTP and HTP were determined using a change-point analysis, and followed the two climatic seasons of Mali: the dry season and the rainy season. Similar results were reported by Sissoko et al. [25] and Coulibaly et al. in Mali [41], by Dieng et al. in Senegal [5], and by Rouamba et al. [33] and Baragatti et al. in Burkina Faso [42].

The distribution of malaria hotspots varied significantly over the study period. Some health areas, most of them in urban areas, remained non-hotspots throughout the study. Others were hotspots in the first year of the study or in two consecutive years, and others were hotspots during every year of the study (Table 2). The non-negligible incidence observed during the LTPs reflected the persistence of malaria during the dry season. This phenomenon, which was also observed in Senegal by Dieng et al. [5], suggests that malaria control strategies need to be implemented earlier than is currently the case.

The CART analysis found a very low risk of malaria transmission in areas with little cropland. This risk was four times higher in areas with large cropland areas and little bare soil, and two times higher in areas with large cropland areas and large bare soil areas (Figure 5). This finding suggests that market gardening, which is widespread in the Kati district, increases the risk of malaria transmission—a phenomenon explained by the fact that the water reservoirs needed to sustain this activity constitute potential mosquito breeding sites. This finding is also in line with the studies conducted in other parts of the world. Thus, Cissoko et al. [21] and Bhattarai et al. [43] found malaria incidence to be associated with land use and the proximity of water sources in Mali and Nepal, respectively. For their part, Mitchell et al. [44] and Paul et al. [45] observed an association between greater cropland cover and increased malaria prevalence in Tanzania. Lastly, Nicholas et al. [46] found a link between agricultural practices and the proliferation of malaria vectors in Kenya. 

The limitation of our study was the use of data from consultation records. Most of these records were not properly maintained. In addition, information was missing in some observations. As a corrective action, all incomplete observations were removed from the database. Another limitation of this study was the absence of the consideration of social and behavioral factors, which are important in the implementation of malaria control interventions.

## 5. Conclusions

Our study confirmed the important variability of malaria incidence in the health district of Kati and confirmed that malaria’s variability was highly associated with a combination of meteorological and environmental factors, including land cover and land use. We found that malaria incidence was low in areas with little cropland areas, but much higher in areas with large cropland areas and little bare soil areas. The persistence of malaria cases during the dry season and the spatio-temporal variability of malaria hotspots reinforce the need for innovative and targeted strategies in Mali.

## Figures and Tables

**Figure 1 ijerph-19-14361-f001:**
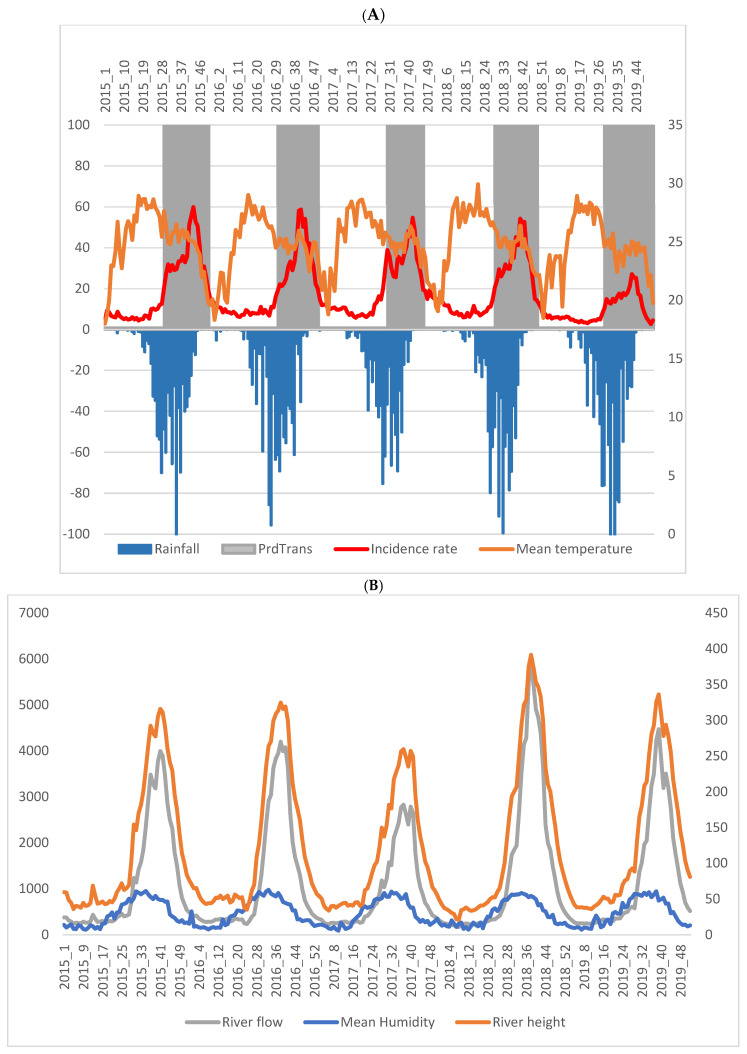
(**A**): Weekly malaria incidence (red curve) according to rainfall (blue), temperature (orange), and transmission period (PrdTrans: white band—low transmission period; grey band—high transmission period). (**B**): River height (orange curve), river flow (gray curve), and relative humidity (blue curve).

**Figure 2 ijerph-19-14361-f002:**
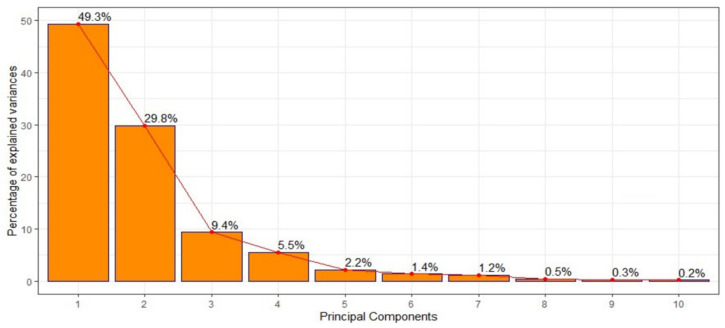
Percentage of inertia, explained by the synthetic indicators in principal component analysis.

**Figure 3 ijerph-19-14361-f003:**
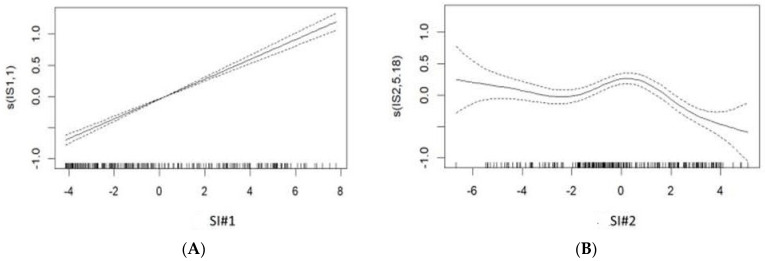
Generalized additive model of synthetic indicators #1 (**A**) and #2 (**B**). The solid line represents the relationship between synthetic indicators #1 or #2 and malaria incidence, and the dotted line represents the 95% confidence interval.

**Figure 4 ijerph-19-14361-f004:**
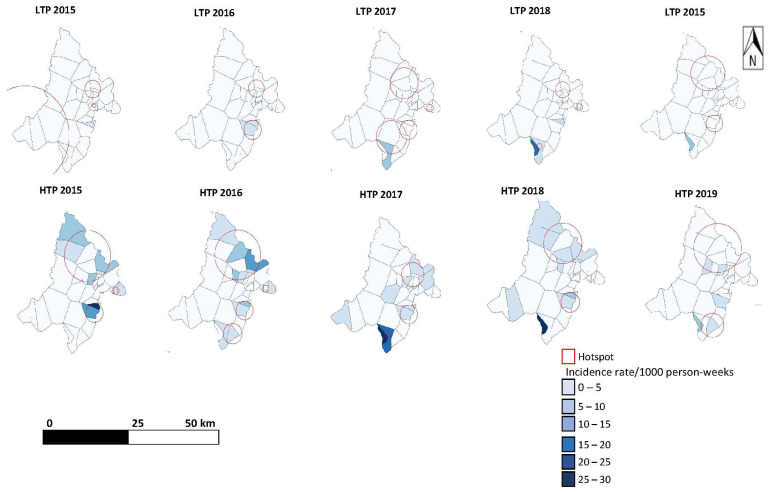
Maps showing the spatio-temporal distribution of malaria incidence and hotspots (red circles) in Kati health district. LTP—low transmission period; HTP—high transmission period.

**Figure 5 ijerph-19-14361-f005:**
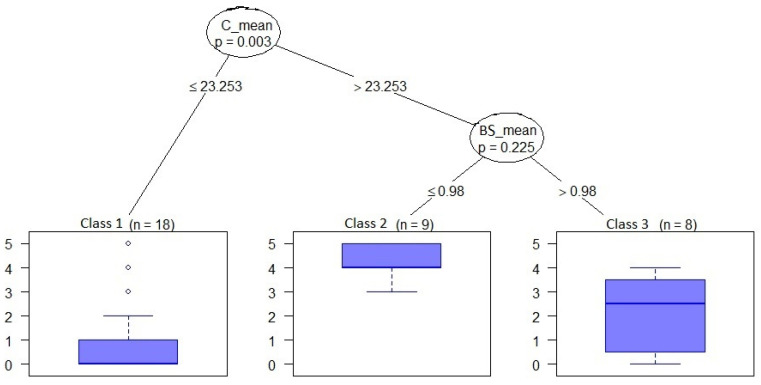
CART classification of health areas, according to the level of malaria risk and to cropland cover and bare soil cover. The blue box represents the number of times a health area was a hotspot. C_mean—mean cropland cover; BS_mean—mean bare soil cover.

**Figure 6 ijerph-19-14361-f006:**
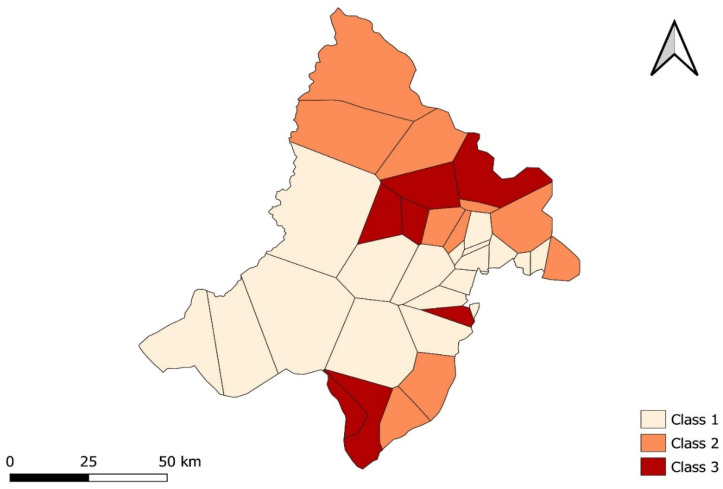
Map of health areas classified according to level of malaria risk. Class 1 corresponds to a low level of malaria risk, Class 2 to a high level of malaria risk, and Class 3 to a moderate level of malaria risk.

**Table 1 ijerph-19-14361-t001:** Correlation of the main synthetic indicators with malaria incidence.

Synthetic Indicators	Variables	Percentage of Inertia	Correlation Coefficient r	Percentage of Total Inertia
**Synthetic Indicator #1**	Mean river flow	7.05	0.87	49.3%
	Maximum river flow	7.12	0.88	
	Mean river height	7.49	0.90	
	Maximum river height	7.50	0.90	
	Minimum river height	7.44	0.90	
	Mean night relative humidity	7.92	0.93	
	Minimum night relative humidity	7.55	0.90	
	Mean day relative humidity	7.89	0.93	
	Maximum day relative humidity	7.71	0.91	
	Minimum day relative humidity	7.69	0.91	
	Cumulative Rainfall	4.47	0.70	
	Wind speed	3.87	−0.65	
**Synthetic Indicator #2**	Mean air pressure	7.61	0.71	29.8%
	Mean night temperature	14.52	0.98	
	Minimum night temperature	13.21	0.93	
	Maximum night temperature	14.24	0.97	
	Mean day temperature	14.52	0.98	
	Minimum day temperature	13.21	0.93	
	Maximum day temperature	14.24	0.97	

**Table 2 ijerph-19-14361-t002:** Distribution of health areas according to hotspot status.

	Never a Hotspot	Hotspot in the First Year of the Study	Hotspot in Two Consecutive Years	Hotspot Every Year of the Study	Hotspot without Temporal Regularity
**Number of health areas**	10	3	2	10	10

**Table 3 ijerph-19-14361-t003:** Transmission periods according to the presence of hotspots and selected meteorological variables.

Malaria Transmission Periods	Duration (Weeks)	Number of Health Areas That Were a Hotspot	Number of Hotspots	Incidence Rate (Cases per 1000 Person–Weeks)			Rainfall (mm)			Mean Temperature (°C)			Mean Wind (km/h)			Mean NDVI		
				Min.	Median	Max.	Min.	Median	Max.	Min.	Median	Max.	Min.	Median	Max.	Min.	Median	Max.
Low transmission period, 2015 (W1–W28)	28	10	5	0.003	0.737	7.210	203	260	305	29	29	31	2.06	2.24	2.34	0.218	0.274	0.344
High transmission period, 2015 (W29–W50)	22	15	5	0.041	2.883	27.2	578	725	853	25	26	27	1.43	1.57	1.70	0.240	0.536	0.646
Low transmission period, 2015 (W51)–2016 (W29)	31	9	7	0.001	0.994	8.742	321	408	455	28	29	30	2.21	2.31	2.39	0.240	0.348	0.440
High transmission period, 2016 (W30–W49)	20	16	5	0.022	2.803	15.75	433	520	603	26	27	28	1.06	1.22	1.34	0.240	0.377	0.430
Low transmission period, 2016 (W50)–2017 (W26)	29	14	6	0.004	1.140	12.02	200	257	307	28	29	30	1.96	2.14	2.25	0.240	0.337	0.451
High transmission period, 2017 (W27–W47)	21	10	6	0.176	3.198	25.04	543	611	679	26	27	28	1.12	1.35	1.50	0.240	0.447	0.599
Low transmission period, 2017 (W48)–2018 (W27)	32	8	5	0.019	1.206	21.34	211	278	326	28	29	30	1.91	2.03	2.15	0.240	0.340	0.400
High transmission period, 2018 (W28–W49)	22	13	7	0.003	2.034	36.44	641	741	842	26	27	27	1.11	1.27	1.40	0.240	0.506	0.616
Low transmission period, 2018 (W50)–2019 (W27)	30	13	7	0.023	0.381	11.72	166	213	242	28	29	30	1.99	2.12	2.19	0.204	0.266	0.301
High transmission period, 2019 (W28–W46)	19	14	4	0.005	0.399	14.29	689	789	906	26	27	27	1.11	1.40	1.56	0.220	0.247	0.269

**Table 4 ijerph-19-14361-t004:** Health areas according to the number of times they were a hotspot and their land cover characteristics.

Health Areas	Number of Times Health Area Was a Hotspot	Bare Soil			Cropland			Grass Vegetation			Shrubland			Forest			Seasonal Inland Water			Permanent Inland Water		
		Min.	Median	Max.	Min.	Median	Max.	Min.	Median	Max.	Min.	Median	Max.	Min.	Median	Max.	Min.	Median	Max.	Min.	Median	Max.
**BANCOUMANA**	2	0	0.33	100	0	37.8	90	0	33.6	100	0	15.8	43	0	7.4	42	0	0	100	0	0	100
**DABAN**	4	0	0	23	0	29	87	0	40.6	81	0	20.8	47	0	7.6	54	0	0	81	0	0	0
**DIAGO**	4	0	0.67	30	0	31.6	87	0	39.4	76	0	17.8	44	0	7.4	39	0	0	6	0	0	0
**DIALAKORODJI**	0	0	0	34	0	15	78	0	39.4	74	0	16.4	43	0	6.4	38	0	0	0	0	0	0
**DIO BA**	4	0	0	20	0	15.4	85	0	42.2	79	0	24.8	47	0	9.8	42	0	0	5	0	0	0
**DIO GARE**	3	0	0.33	26	0	24.2	86	0	39.4	77	0	21.6	44	0	8.8	40	0	0	25	0	0	0
**DJIGUIDALA**	3	0	0	9	0	20.8	88	0	42	100	0	23.8	46	0	10.6	46	0	0	100	0	0	100
**DJOLIBA**	0	0	0	100	0	33.4	93	0	35	99	0	17.8	66	0	7.4	40	0	0	100	0	0	100
**DOGODOUMA**	2	0	0	24	0	6.8	39	0	43.8	73	0	24	40	0	15.6	48	0	0	0	0	0	0
**DOMBILA**	0	0	0	29	0	4.2	87	0	45.8	80	0	25.8	43	0	14	55	0	0	11	0	0	0
**DOUBABOUGOU**	0	0	0	16	0	10	82	0	44.2	78	0	25.4	43	0	11.8	46	0	0	2	0	0	0
**FALADJE**	4	0	0	26	0	21.6	86	0	42.8	84	0	23.8	45	0	8.8	53	0	0	70	0	0	0
**FARABANA**	5	0	0	69	0	6.4	82	0	41.2	99	0	24.4	46	0	13	62	0	0	100	0	0	100
**FARADA**	0	0	0.67	36	0	27.8	80	0	37.4	72	0	17.6	40	0	7.6	38	0	0	3	0	0	0
**KABALABOUGOU**	0	0	0	89	0	0	84	0	17.4	100	0	2.7	39	0	0	32	0	0	100	0	0	100
**KALIFABOUGOU**	5	0	0	22	0	23	86	0	42.2	79	0	23.2	46	0	8.8	45	0	0	7	0	0	0
**KANADJIGUILA**	0	0	0	28	0	3.8	79	0	45	75	0	25.4	41	0	13.2	43	0	0	0	0	0	0
**KATICORO**	3	0	0.33	16	0	18	79	0	40.4	74	0	15.2	36	0	5.8	33	0	0	0	0	0	0
**KOKO**	0	0	0	25	0	17.4	80	0	41.6	78	0	21.4	46	0	9.8	39	0	0	0	0	0	0
**MALIBOUGOU**	0	0	0.33	22	0	21.6	79	0	36.2	67	0	19	41	0	10	46	0	0	53	0	0	0
**MORIBABOUGOU**	4	0	0	94	0	4.4	72	0	42	100	0	22.2	45	0	8.6	41	0	0	100	0	0	63
**NANA-KENIEBA**	0	0	0	17	0	0	80	0	47.8	81	0	27.2	46	0	22.4	56	0	0	0	0	0	0
**NEGUELA**		0	0	30	0	5.4	87	0	42.2	80	0	27.2	50	0	11.8	64	0	0	17	0	0	0
**NGABAKORO**	3	0	0.33	59	0	23.2	85	0	39.8	100	0	18.4	66	0	7.6	43	0	0	100	0	0	100
**NIAME**	3	0	0	13	0	15.6	80	0	45.4	75	0	25.6	47	0	10.6	42	0	0	0	0	0	0
**NIOUMA MAKANA**	1	0	0	22	0	0.8	80	0	49.8	81	0	26.4	45	0	19	61	0	0	0	0	0	0
**OUEZZINDOUGOU**	4	0	0	63	0	19.8	83	0	31.8	87	0	19.8	42	0	9.2	55	0	0	100	0	0	100
**SAFO**	1	0	0	19	0	25.6	81	0	44.2	78	0	20.6	49	0	7	38	0	0	5	0	0	0
**SANANFARA**	0	0	0	37	0	10.2	43	0	43	77	0	22	42	0	8.4	49	0	0	0	0	0	0
**SANDAMA**	0	0	0	24	0	0	80	0	48.4	81	0	27.2	45	0	20.8	60	0	0	0	0	0	0
**SANGAREBOUGOU**	0	0	0	98	0	3.4	69	0	31	97	0	8.2	46	0	4	36	0	0	100	0	0	15.6
**SIBY**	0	0	0	28	0	4.4	85	0	43.6	79	0	23.8	46	0	13.2	53	0	0	75	0	0	0
**SONIKEGNY**	5	0	0	19	0	29.8	66	0	40.6	78	0	19	39	0	7.6	34	0	0	47	0	0	0
**TORODO**	4	0	0	32	0	27	86	0	41	75	0	22.2	47	0	8.2	43	0	0	26	0	0	0
**YELEKEBOUGOU**	5	0	0	28	0	25.6	85	0	43	79	0	22.4	46	0	8.2	43	0	0	55	0	0	0

**Table 5 ijerph-19-14361-t005:** Classes of malaria risk according to the number of health areas and to meteorological and environmental variables.

Class	1	2	3
**Incidence (cases per 1000 persons–weeks)**	288.62	425.89	262.80
**Number of health areas**	18	9	8
**Total Population**	112,979	56,851	83,618
**Cumulative Rainfall (mm)**	992	902	950
**Mean Temperature (°C)**	27.78	28.24	27.95
**Mean NDVI**	0.37	0.35	0.33
**Mean Cropland (SD)**	17.68 (2.67)	29.83 (2.36)	29.88 (3.72)
**Mean Shrubland (SD)**	19.93 (2.13)	21.55 (0.81)	18.09 (1.40)
**Mean Forest (SD)**	12.82 (1.75)	10.07 (1.56)	9.22 (2.31)
**Mean Permanent Inland Water (SD)**	1.50 (9.67)	0.04 (1.77)	1.84 (9.56)
**Mean Seasonal Inland Water (SD)**	1.32 (8.82)	0.04 (1.14)	1.46 (7.84)
**Mean Bare Soil (SD)**	1.50 (2.45)	0.60 (0.43)	1.62 (1.35)
**Mean Grass Vegetation (SD)**	35.83 (3.57)	36.87 (0.89)	34.27 (2.31)

NDVI—normalized difference vegetation index; SD—standard deviation. Class 1 corresponds to a low level of malaria risk, Class 2 to a high level of malaria risk, and Class 3 to a moderate level of malaria risk.

## Data Availability

Data can be shared as needed.

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
