# Peer review of "Spatio-Temporal Variability of Malaria Incidence in the Health District of Kati, Mali, 2015–2019"

_ijerph, 2022, doi:10.3390/ijerph192114361_

Round 1

Reviewer 1 Report

I have no serious comments on the peer-reviewed article, but, nevertheless, I would like to make a few minor comments:

1) Line 55. I think at this point the authors could decipher "plasmodium index", it will be convenient for a reader who is not familiar with malaria

2) Lines 85-86 It is not entirely clear why the indicator is more than 100%

3) Figure 1 Panel A  In the caption in the figure "Incidence rate_" the "_" sign is superfluous

4) Line 231 Text formatting error, "Figure" printed at the far right, the number of this Figure moved to the next line

5) Table 4. - in the version that was given to me, the right side of the table is cut off, in the row "SANGAREBOUGOU" the cell border lines are thinner than in the rest of the table; there is no line between "DOUBABOUGOU" and "FALADJE" in "Forest Max" column

6) Line 353 ")" should be replaced with "]"

7) Lines 373-374 "The incidence of malaria increased significantly as temperature increased but decreased significantly at higher temperatures, a phenomenon explained by the fact that larvae and adults die at temperatures above 35°C"

8) It seems to me that it is worth changing this phrase a little and clarifying exactly whose larvae and adults are meant.

Reviewer 2 Report

It focuses on spatial and temporal aspects. In the collection of data it says that there are variables such as age, sex, weight, etc. that they do not analyze later. They should take into account social factors. For the analysis of malaria, the space must be seen from the social perspective. If you do not want to analyze them and keep the article present, let them say so in the conclusions as one of the shortcomings of the study.

The conclusions are too short. You have to expose in more detail the findings found.
